# The '*Candidatus* Phytoplasma mali' effector protein SAP11$_{CaPm}$ interacts with MdTCP16, a class II CYC/TB1 transcription factor that is highly expressed during phytoplasma infection

**Cecilia Mittelberger**[1], **Bettina Hause**[2], **Katrin Janik**[1]*

**1** Molecular Biology and Microbiology, Group of Functional Genomics, Research Centre Laimburg, Pfatten (Vadena), South Tyrol, Italy, **2** Department of Cell and Metabolic Biology, Leibniz Institute of Plant Biochemistry, Halle, Saxony-Anhalt, Germany

* katrin.janik@laimburg.it

**Data Availability Statement:** All relevant data are within the article and its Supporting Information files.

## Abstract

'*Candidatus* Phytoplasma mali', is a bacterial pathogen associated with the so-called apple proliferation disease in *Malus × domestica*. The pathogen manipulates its host with a set of effector proteins, among them SAP11$_{CaPm}$, which shares similarity to SAP11$_{AYWB}$ from '*Candidatus* Phytoplasma asteris'. SAP11$_{AYWB}$ interacts and destabilizes the class II CIN transcription factors of *Arabidopsis thaliana*, namely AtTCP4 and AtTCP13 as well as the class II CYC/TB1 transcription factor AtTCP18, also known as BRANCHED1 being an important factor for shoot branching. It has been shown that SAP11$_{CaPm}$ interacts with the *Malus × domestica* orthologues of AtTCP4 (MdTCP25) and AtTCP13 (MdTCP24), but an interaction with MdTCP16, the orthologue of AtTCP18, has never been proven. The aim of this study was to investigate this potential interaction and close a knowledge gap regarding the function of SAP11$_{CaPm}$. A Yeast two-hybrid test and Bimolecular Fluorescence Complementation *in planta* revealed that SAP11$_{CaPm}$ interacts with MdTCP16. MdTCP16 is known to play a role in the control of the seasonal growth of perennial plants and an increase of *MdTCP16* gene expression has been detected in apple leaves in autumn. In addition to this, *MdTCP16* is highly expressed during phytoplasma infection. Binding of MdTCP16 by SAP11$_{CaPm}$ might lead to the induction of shoot proliferation and early bud break, both of which are characteristic symptoms of apple proliferation disease.

## Introduction

'*Candidatus* Phytoplasma mali' ('*Ca*. P. mali') is the bacterial pathogen associated with the so-called apple proliferation disease [1]. Apple trees affected by this disease show different growth aberrations like uncontrolled shoot proliferation or enlarged stipules, early bud break and flowering and so-called witches' brooms. Symptomatic trees produce small, tasteless, and

**Funding:** The work of CM and KJ was co-funded by the Autonomous Province of Bozen/Bolzano (Italy, https://www.provinz.bz.it) and the South Tyrolean Apple Consortium (Italy, https://www.apfelwelt.it). The study was realized within the APPLIII and APPLIV project belonging to the Framework agreement in the field of invasive species in fruit growing and major pathologies (PROT. VZL_BZ 09.05.2018 0002552). The funders had no role in study design, data collection and analysis, decision to publish, or preparation of the manuscript.

**Competing interests:** The authors have declared that no competing interests exist.

unmarketable fruits, which causes economic losses and threatens apple production [2]. Phytoplasmas are wall-less, phloem-limited bacteria, that need insect vectors for their transmission and use effector proteins to manipulate their host plants [3–6]. 'Ca. P. mali' is transmitted by certain psyllids [7] and encodes several potential effector proteins that appear to be released into *Malus* plants by a Sec-dependent secretion system [8,9]. So far, the best characterized effector protein is SAP11$_{CaPm}$, which shows similarity to SAP11$_{AYWB}$ from 'Candidatus Phytoplasma asteris' that is associated with Aster Yellow Witches' Broom disease in aster [10]. Despite of being the best described effector, SAP11$_{CaPm}$'s function is not fully unraveled yet, and there are still knowledge gaps regarding its interaction partners in *Malus × domestica*, the natural host of 'Ca. P. mali'.

The development of witches' brooms, i.e., uncontrolled branching, is a common symptom not only for apple proliferation, but also for several other phytoplasma diseases. Overexpression of SAP11$_{AYWB}$ and SAP11-like proteins from other phytoplasma species resulted in an increased formation of lateral shoots [11–15]. Interestingly, SAP11$_{AYWB}$, SAP11$_{CaPm}$ and other SAP11-like proteins bind and destabilize different TEOSINTE BRANCHED1/CYCLOIDEA/PROLIFERATING CELL FACTOR 1 and 2 (TCP) transcription factors [12,14,16].

TCP transcription factors are highly conserved in all land plant lineages and contain a basic helix-loop-helix (bHLH) domain, responsible for DNA binding [17]. TCPs are classified according to sequence differences in the bHLH-domain in class I and class II subfamilies [18]. The class II TCPs are further subdivided in CINCINNATA (CIN)-like TCPs and in CYCLOIDEA/TEOSINTE BRANCHED1 (CYC/TB1)-like TCPs [19]. TCPs of both classes play a key role in morphological development of plants, stress adaptions and plant immunity [20] and are thus interesting targets of diverse pathogen effector proteins [21,22].

A total of 52 TCP-domain containing genes were identified in apple [23]. For CIN-like class II TCPs, MdTCP25 (orthologue to AtTCP4) and MdTCP24 (orthologue to AtTCP13), an interaction with SAP11$_{CaPm}$ has been shown [8]. Additionally, a yeast-two-hybrid (Y2H) screen using SAP11$_{CaPm}$ revealed cDNA fragments of MdTCP16, a AtTCP18-like CYC/TB1 class II TCP and of the putative chlorophyllide b reductase NYC1 as putative interaction partners [8]. An interaction between SAP11$_{CaPm}$ and the corresponding full-length gene products of the MdTCP16- and MdNYC1-fragment could not be confirmed at that time. Shortly after, the genome of *Malus × domestica* was *de novo* assembled and updated [24]. This sheds new light on the full-length ORFs including *MdTCP* encoding genes and led to some new sequence and reading-frame information regarding the assigned full-length genes of *MdTCP16* and *MdNYC1*.

AtTCP18 is a key regulator of shoot branching [25–27] and is also known as BRANCHED1 (BRC1). BRC1 is used as a common term for all AtTCP18 orthologues in other plant species. Its expression is repressed by cytokinin, gibberellin, phytochrome B and sugar, and is promoted by auxin, an important regulator of apical dominance, strigolactones and a low red to far-red light ratio [28]. BRC1 influences not only the plant architecture, but also the seasonal growth of perennial plants, such as temperate fruit trees by responding to photoperiodic changes [29]. In addition, BRC1 interacts with the FLOWERING LOCUS T protein in Arabidopsis and represses floral transition in axillary meristems [30].

Since several other SAP11-like proteins interact with AtTCP18-like TCPs [11,13,15,31,32] and SAP11$_{CaPm}$ from 'Ca. P. mali' strain PM19 showed an interaction with AtTCP18 in an Y2H screen [16], the aim of this study was a detailed analysis of the two potential interactions between SAP11$_{CaPm}$ with either MdTCP16 (i.e. the orthologue of AtTCP18) or MdNYC1. Moreover, to gain a better understanding of the role of MdTCP16, MdTCP24 and MdTCP25 in the host plant during phytoplasma-infection, the expression of these *TCPs* was determined by qPCR analysis of non-infected and infected *Malus × domestica* leaf samples in spring and autumn.

## Materials and methods

### Yeast-two-hybrid screen

In a previous study cDNA sequences that were partially similar to genes encoding *MdTCP16* and *MdNYC1* were identified as interactors of SAP11$_{CaPm}$ from 'Ca. P. mali' strain STAA (Accession: KM501063) by a Y2H screen [8]. These cDNA nucleotide sequences were obtained by sequencing the prey vectors in the identified yeast colonies and blasting against the NCBI nt-database [33,34]. The sequence, identified as *MdTCP16*, shared 100% sequence identity with the reference sequence XM_008376500.2 and covers 69.5% of the coding sequence for protein XP_008374722.1 (S1 Fig). Primers were designed to amplify the coding sequence of XP_008374722.1 (S1 Table). These primers contained overhangs that attached *Sfi*I-restriction sites to the amplicon for subsequent cloning into the pGAD-HA Y2H prey vector.

The sequence identified as *MdNYC1* shared 100% sequence identity to the reference sequence XM_029109831.1 and covers 13.7% of the C-terminal end of XP_028965664.1. Primers were designed to amplify the coding sequence of XP_028965664.1 and of the C-terminal part identified as an interactor of SAP11$_{CaPm}$ in the Y2H screen. Both amplicons were subcloned into pGAD-HA prey vector via their primer-attached *Sfi*I-overhangs.

The Y2H prey vector was co-transformed with pLexA-N-SAP11$_{CaPm}$ bait vector into the *Saccharomyces cerevisiae* strain NMY51 cells [8,35]. Growth was monitored four days after transformation on selective SD plates lacking the amino acids adenine, leucine, histidine and tryptophane.

### Bimolecular fluorescence complementation analysis

Primers, specific for *MdTCP16* and *SAP11$_{CaPm}$* from 'Ca. P. mali' strain STAA and with *attB*-site overhangs were designed (S1 Table) and fragments were amplified in a total of 50 μL reaction volume using 10 ng template and a final concentration of 1 x iProof HF Buffer, 200 μM dNTPs (50 μM each nucleotide), 0.5 μM of each primer and 0.02 U/μl iProof DNA Polymerase (Bio-Rad, Hercules, CA). The cycling conditions were 30 sec of initial denaturation at 98˚C followed by 35 cycles of 98˚C for 10 sec, 60˚C for 30 sec and 72˚C for 90 sec. The final extension was carried out at 72˚C for 5 min. The amplified fragments were analyzed on a 1.2% agarose-gel and extracted with Kit Montage Gel Extraction columns (Millipore, Bedford, MA). The purified amplicons were used in a BP-reaction for the creation of Gateway-Entry vectors using 1 μL purified amplicon, 100 ng of donor vector (pDONR221-P1P4 or pDONR221-P3P2), 1 μL BP Clonase™ II enzyme (Invitrogen, Carlsbad, CA) following the manufacturer's protocol. Depending on the N- or C-terminal location of the split-YFP in pBiFC vectors [36], *MdTCP16* and *SAP11$_{CaPm}$* entry-vectors with or without stop-codon were combined in a LR-reaction using Gateway™ using LR Clonase™ II enzyme mix (Invitrogen) and following the manufacturer's instructions.

*Nicotiana benthamiana* leaf mesophyll protoplasts were isolated for bimolecular fluorescence complementation analysis (BiFC) from leaves of four-week-old plants and protoplasts were transformed with the pBiFC vectors as described in Janik et al. (2017) [37]. In detail, leaf pieces of two *N. benthamiana* leaves were vacuum infiltrated for 30 min with 10 mL of enzyme solution containing 0.4 M mannitol, 20 mM KCl, 20 mM 4-morpholineethanesulfonic acid (MES), 1.5% (w/v) cellulase R-10, 0.4% (w/v) macerozyme R-10, 10 mM CaCl$_2$, and 1 mg/mL bovine serum albumin. The leaf pieces were then incubated in the dark at room temperature for 4 h and afterwards gently shaken for 30 min in the dark to release the protoplasts. The protoplast-leaf pieces solution was filtrated through a cell strainer into two 15 mL round bottom centrifuge tubes and centrifuged at 200 x g and 4˚C for 1 min. The supernatant was carefully removed, and the first pellet was resuspended in 3 mL buffer W5 (154 mM NaCl, 125 mM CaCl$_2$, 5 mM KCl, 2 mM MES at pH 5.7). The second pellet was carefully resuspended with

the first suspension. The protoplast suspension was incubated in the dark on ice for 40 min for sedimentation of the protoplasts. The protoplast pellet was again carefully resuspended in 3 mL buffer W5 and incubated for another 40 min in the dark on ice. The protoplast pellet was resuspended in 2 mL buffer MMG (0.4 M mannitol, 15 mM $MgCl_2$, 4 mM MES at pH 5.7) and protoplast density was counted in a Neubauer improved cell counting chamber. The protoplast solution was diluted with buffer MMG to a final concentration of 100,000 protoplasts per mL. 10 µg of pBiFC vector plasmid DNA, obtained by plasmid preparation with NucleoSnap Plasmid Midi kit (Macherey-Nagel, Düren, Germany) followed by a PEG precipitation, were mixed with 200 µL of protoplast solution. 220 µL of PEG-transformation-solution (0.2 M mannitol, 0.1 M $CaCl_2$, 40% (w/v) PEG 4000) were added to the protoplast-DNA mix and mixed by carefully shaking the tube. After 5 min of incubation at room temperature, 880 µL of buffer W5 were carefully added to the suspension. The protoplast transformation solution was centrifuged for 1 min at 4°C and the pellet was resuspended in 200 µL buffer WI (0.5M mannitol, 20mM KCl, 4mM MES). The tubes were then placed horizontally in the dark for incubation at room temperature overnight. Transformed protoplasts were analyzed 16h after transformation using a confocal laser scanning microscope (Zeiss LSM800, Carl Zeiss Microscopy, Oberkochen, Germany). Transformation rate and BiFC rate were determined as a mean value of three independent repetitions of protoplast transformation.

## Plant material

Trees in the greenhouse: Apple rootstocks M9 were graft inoculated with 'Ca. P. mali' strain PM6 (AT2-subtype) [38] infected or non-infected control scions of *Malus x domestica* cultivar (cv.) Golden Delicious and kept in greenhouse without temperature or light control. These small, one-year-old plants had a maximum height of about 40 cm.

Trees in the foil tunnel: Fully grown, ca. 2 m high, naturally infected *Malus x domestica* cv. Golden Delicious trees were grown and kept in an insect safe foil tunnel. Trees were infected with the locally predominant AT2 strain as determined by Sanger sequencing. Since it was a natural infection it cannot be ruled out, however, that a mixed infection with other strains was present in these trees. Nevertheless, this situation resembles the one in the field. Leaf samples for the analyses were taken in May and October 2011 as described in Janik et al. 2017 [8]: For each time-point pools of leaves from non-infected and infected trees (eleven trees/pool in May and six trees/pool in October), respectively, were assembled. Each of the pools of infected trees comprised material from trees representing the same symptom intensities. Equal amounts of material from each tree were pooled.

The phytoplasma levels from pooled leaf and root samples are reported in S2 Fig.

## RNA extraction and cDNA synthesis

Greenhouse samples were collected in May and October 2021 from single apple trees. Leaf discs from five non-infected and five infected, one year old apple trees, grown in greenhouse were excised, and immediately frozen in liquid nitrogen. Leaf discs were grinded with mortar and pestle under liquid nitrogen. 100 mg of leaf powder was used for RNA extraction following protocol A of the Spectrum™ Plant Total RNA Kit (Sigma-Aldrich, St. Louis, MO). RNA was eluted in 50 µL elution buffer. RNA concentration was measured with a Spectrophotometer (NanoPhotometer® N60, Implen, München, Germany). 1 µg or 2 µg of RNA were reverse transcribed into cDNA by using SuperScript™ IV VILO™ Master Mix with ezDNase enzyme (Invitrogen), following the protocol that includes gDNA digestion with DNase enzyme. Samples that contained all reagents and RNA except the reverse transcriptase ("No-RT controls") were performed in parallel. Synthesized cDNA was stored at -80°C.

Sampling of naturally infected trees of *Malus* × *domestica* cv. Golden Delicious was done in May and October 2011. Three pools of leaves from control or 'Ca. P. mali'-infected trees (6–11 trees/pool) were tested. RNA extraction and cDNA preparation is described in Supplementary Material and Methods of Janik et al. (2017) [8]. Synthesized cDNA was stored at -80˚C.

### DNA extraction

DNA of *Malus* × *domestica* leaf samples was extracted from approx. 50 mg grinded leaf tissue, using the DNeasy Plant Mini Kit (Qiagen, Venlo, The Netherlands), following the manufacturer's instruction. DNA was eluted in 50 μL elution buffer.

### qPCR analysis

The transcription factors *MdTCP16*, *MdTCP25*, *MdTCP24* and the effector protein $SAP11_{CaPm}$ were amplified with specific qPCR primer pairs (S1 Table) using a SYBR-Green qPCR assay. Reactions were run in a total of 10 μl, using 2 x Universal KAPA SYBR® FAST master mix (KAPABIOSYSTEMS, Wilmington, MA), 20 pmol of forward and reverse primer and 2 μl of 1:50 diluted cDNA samples. Additionally, qPCR master mixes for the reference genes *GAPDH*, *tip41* and *EF1α* were prepared for all samples, using the same reagent and template concentrations, and cycling conditions as mentioned above. All targets were amplified on the same 384well plate in one qPCR run on a CFX384 Touch Real-Time PCR Detection System (Bio-Rad), using three technical replicates per sample. Cycling conditions were as follows: 95˚C for 20 sec, followed by 35 cycles with 95˚C for 3 sec and 60˚C for 30 sec, melt curve from 65˚C to 95˚C with an increment of 0.5˚C/5 sec.

For determining the qPCR efficiency of every primer-combination, a four-point dilution series (1:10, 1:50, 1:100, 1:200) of a cDNA sample mixture was analyzed in each qPCR run.

Phytoplasma quantity was determined by the detection of the phytoplasma specific *16S* gene together with the *Malus* specific single copy gene *ACO* as described by Baric et al. (2011) [39]. In brief, 2 μl of template DNA were analyzed in a total multiplex reaction volume of 20 μl, using 2x iQ™ supermix master mix (Bio-Rad), 18 pmol of each qAP-16S forward and reverse primer, 4 pmol qAP-16S probe, 4 pmol of each qMD-ACO forward and reverse primer and 4 pmol of qMD-ACO probe. The 'Ca. P. mali' specific qAP-16S probe was 5'-labeled with the reporter dye FAM, while the *Malus* specific qMD-ACO probe was 5'-labeled with VIC. Cycling conditions were as follows 95˚C for 3 min followed by 35 cycles of 95˚C for 15 sec and 60˚C for 60 sec. Reactions were run in triplicates on a CFX96 Touch Real-Time PCR Detection System (Bio-Rad). For determining the qPCR efficiency, a five-point dilution series (undiluted, 1:10, 1:50, 1:100, 1:200) of a sample mixture of DNA from infected *Malus* × *domestica* roots was analyzed in each qPCR run.

### Data analysis

Normalized expression was calculated according to Taylor et al. (2019) [40] considering qPCR efficiency (E) of each run, following the formula $1+E^{\Delta Cq}$ for relative quantity. Statistical analysis was performed with GraphPad Prism 7.05 (GraphPad Software Inc., La Jolla, CA).

## Results

### Yeast-two-hybrid and bimolecular fluorescence complementation analyses

The Y2H test using $SAP11_{CaPm}$ from 'Ca. P. mali' strain STAA fused to the GAL4-binding domain as bait revealed an interaction between MdTCP16 and $SAP11_{CaPm}$. An interaction

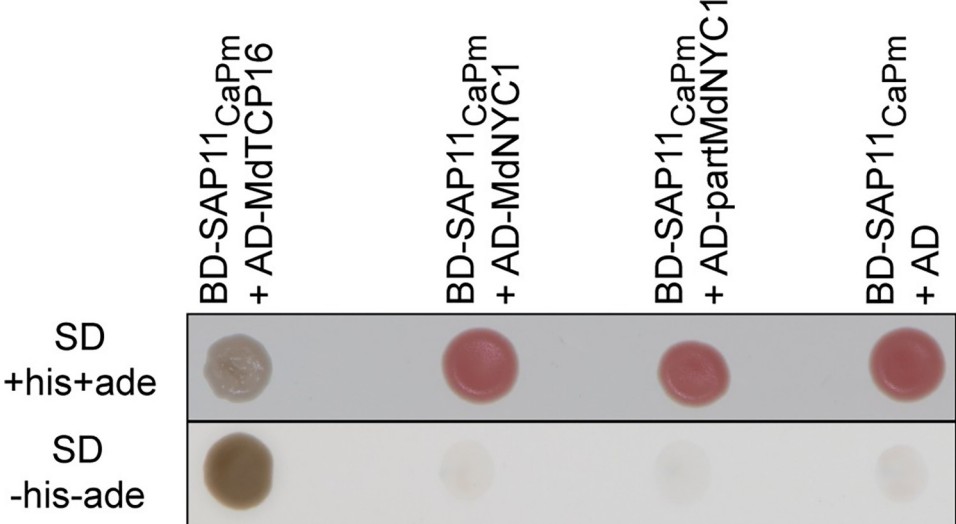

**Fig 1. SAP11$_{CaPm}$ interacts with MdTCP16 but not with MdNYC1 in yeast.** Interaction between the bait SAP11$_{CaPm}$ fused to a DNA binding domain (BD) and the prey protein from *Malus × domestica* fused to an activation domain (AD) is indicated by growth of the *Saccharomyces cerevisiae* reporter strain NMY51 on SD minimal medium lacking the amino acids histidine (his) and adenine (ade). The white color of BD-SAP11$_{CaPm}$ + AD-MdTCP16 grown on full medium is an additional indication for a strong interaction, since the ADE2 reporter gene is activated upon interaction, while in absence of a protein-protein interaction and thus no ADE2 activation, a red colored intermediate accumulates in the adenine metabolic pathway.

between SAP11$_{CaPm}$ and MdNYC1 (neither the fragment nor the full-length protein) was not detectable (Fig 1).

For further confirmation of the interaction between SAP11$_{CaPm}$ and MdTCP16, the sequences of SAP11$_{CaPm}$ and MdTCP16 were subcloned into different pBiFC-2in1 vectors [36] by Gateway-cloning [41]. The pBiFC-2in1 vectors were transformed into *N. benthamiana* mesophyll protoplasts [37] and interaction was analyzed by confocal microscopy 16 h after transformation. Up to 92.5% of the transformed protoplasts showed a YFP signal, resulting from the interaction between SAP11$_{CaPm}$ and MdTCP16 (Fig 2). The YFP-signal was localized

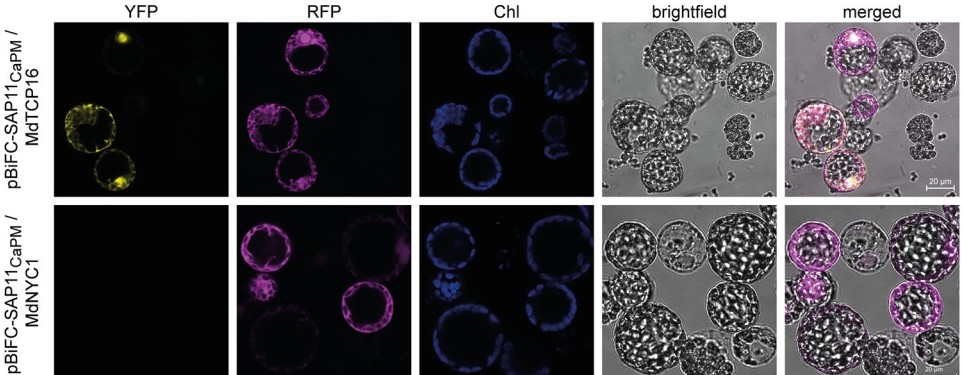

**Fig 2. SAP11$_{CaPm}$ interacts with MdTCP16 *in planta*.** *Nicotiana benthamiana* mesophyll protoplasts were co-transformed with a BiFC expression vector encoding SAP11$_{CaPm}$ and MdTCP16. SAP11$_{CaPm}$ and MdTCP16 interact in the nucleus and in the cytoplasm as indicated by the occurrence of a YFP signal in these two cellular compartments. The co-expression of SAP11$_{CaPm}$ and MdNYC1 did not show any YFP signal. The RFP signal (depicted in magenta) indicates a successful transformation of protoplasts with the BiFC vector. Chl (depicted in blue) shows the autofluorescence of chlorophyll within the chloroplasts. Microscopic analysis was performed with a Zeiss LSM 800 confocal microscope. Bars represent 20 μm for all micrographs.

to the cell nucleus and occurred additionally in the cytoplasm. The co-expression of SAP11-$_{CaPm}$ and MdNYC1 fused to both YFP-halves did not reveal YFP fluorescence after transformation of protoplasts, confirming the Y2H results (Fig 2).

Taken together, these results show that SAP11$_{CaPM}$ interacts with MdTCP16 in addition to the previously shown interaction with MdTCP24 and MdTCP25 [8].

## Expression of TCPs in *Malus* × *domestica* during infection

It was unclear how an infection with 'Ca. P. mali' affects the expression of the three TCP-encoding genes being the targets of its effector SAP11$_{CaPm}$. Thus, to analyze the expression of *MdTCP16*, *MdTCP24* and *MdTCP25* during infection in *Malus* × *domestica* host plant, qPCR assays were used to determine expression levels of the respective TCPs and the effector protein SAP11$_{CaPm}$ (Fig 3).

*MdTCP24* was stably expressed throughout the season and its transcript levels did not change upon infection, regardless of whether the trees were grown in the field or in the greenhouse (Fig 3A and 3B). Also, *MdTCP25* in leaves from greenhouse plants was stably expressed throughout the season, but in naturally infected samples the expression was significantly higher in samples taken in spring than in those taken later in the season, for both non-infected and infected trees. Furthermore, regarding field-grown trees in spring, *MdTCP25* expression is significantly lower in naturally infected samples compared to non-infected samples (Fig 3A and 3B). The expression of *MdTCP16* was lower than those of *MdTCP24* and *MdTCP25*. *MdTCP16* transcript levels were low in both infected and non-infected samples from spring and higher in samples from autumn. Furthermore, it was tendentially, but not significantly higher in infected autumn samples than in non-infected autumn samples (Fig 3A and 3B). SAP11$_{CaPm}$ expression could not be detected in naturally infected spring samples, but it was detectable in the leaves from greenhouse samples throughout the season (Fig 3C–3E). SAP11-$_{CaPm}$ expression in leaves tends to be increased when phytoplasma level in the same tissue is high (Fig 3F).

The expression of *MdTCP16* significantly correlated positively with the expression of SAP11$_{CaPm}$ in leaves from trees of the greenhouse ($R^2 = 0.40$) and there was a trend of this finding also in leaves from trees cultivated in the foil tunnel ($R^2 = 0.52$). *MdTCP16* expression also strongly correlated positively ($R^2 = 0.97$) with the phytoplasma quantity in leaf samples (Fig 3C and 3F, Table 1). *MdTCP16* expression furthermore negatively correlates ($R^2 = 0.74$) with the phytoplasma concentration (3F, Table 1). This is opposite to the findings for *MdTCP25*, whose expression correlated negatively with SAP11$_{CaPm}$ quantity in leaf samples of trees from the greenhouse ($R^2 = 0.15$) and from the foil tunnel ($R^2 = 0.78$) (Fig 3D). The negative correlation between *MdTCP25* and SAP11$_{CaPm}$ expression was significant in leaves from the foil tunnel, whereas in the samples from greenhouse tree a trend of a negative correlation was observed (Fig 3D). *MdTCP24*, in contrast, did neither correlate with the expression of SAP11$_{CaPm}$ nor with the phytoplasma quantity (Fig 3E and 3F).

## Discussion

It is known that SAP11$_{AYWB}$ from AYWB-phytoplasma interacts and destabilizes AtTCP18 in *Arabidopsis thaliana* [12], but it was so far unknown if SAP11$_{CaPm}$ from 'Ca. P. mali' can interact with MdTCP16, the orthologous of AtTCP18 in *Malus* × *domestica*, the actual host plant of this phytoplasma species. It has been shown previously that SAP11$_{CaPm}$ from strain PM19, that shares 99.2% sequence identity to SAP11$_{CaPm}$ from strain STAA used in this study, interacts with AtTCP18 in a Y2H test [16]. However, to understand the potential effects of the effector protein SAP11$_{CaPm}$, it is important to unravel the targets not only in model plants but in the

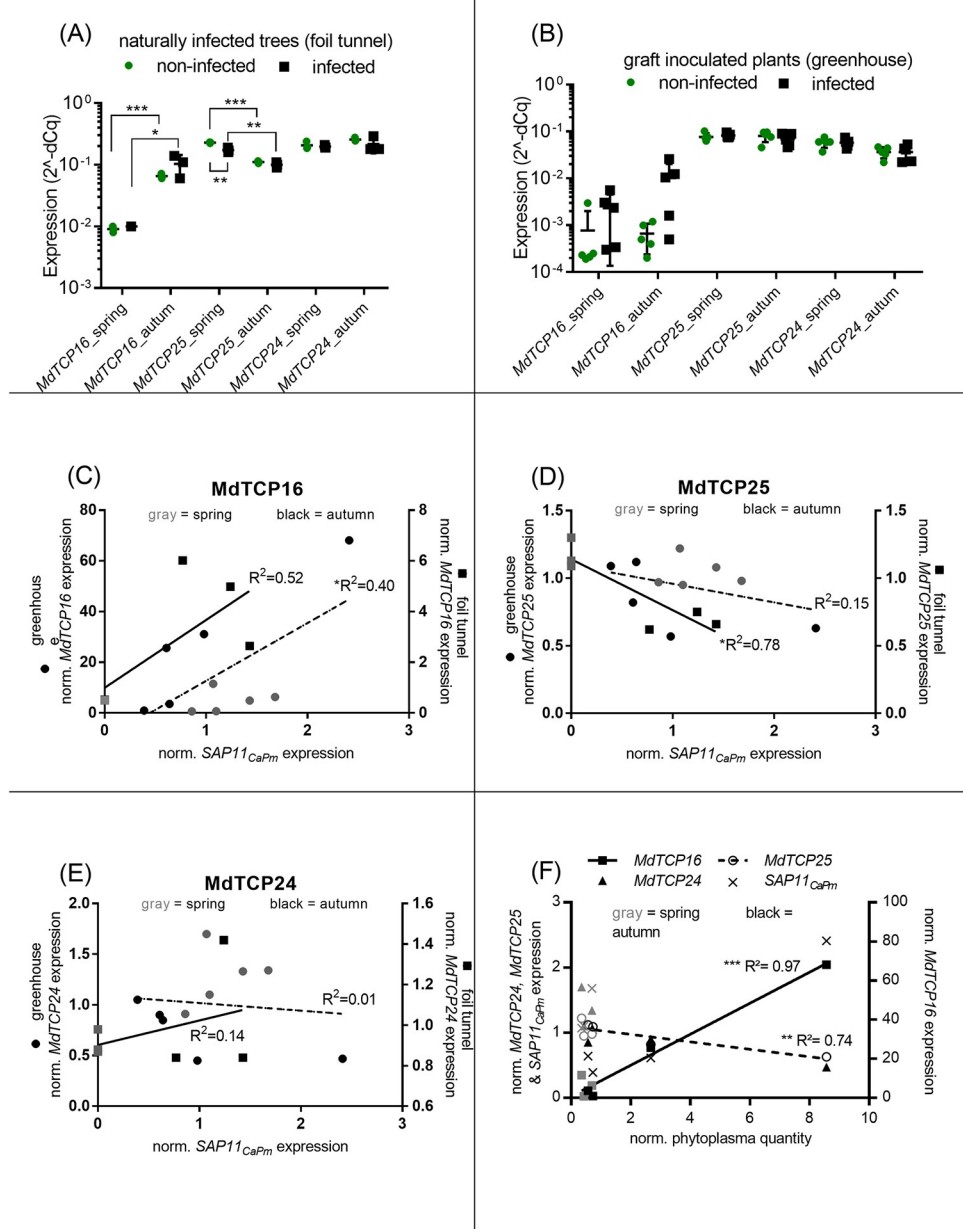

**Fig 3. *MdTCP16* expression increases from spring to autumn, is slightly higher in infected samples and strongly correlates to phytoplasma levels in leaves from infected *Malus* × *domestica*.** *MdTCP* expression in spring and autumn of non-infected (green) and naturally infected (black) apple leaves (A) or graft inoculated and grown in greenhouse plants (B). Naturally infected samples (A) comprise one leaf pool from non-infected trees, and three leaf pools from infected trees. Graft inoculated greenhouse plants (B) comprise leaf samples from 5 non-infected and 5 infected trees, each represented by a data point. Correlation of *MdTCP* and *SAP11CaPm* expression (C, D, E) in infected *Malus* × *domestica* leaf-samples from spring (grey, n = 8) and autumn (black, n = 8). Lines in graphs C, D, E and F show linear regression of the respective samples either from the greenhouse (dashed line) or from the foil tunnel (solid line). Samples in C, D and E were grouped regarding their growing conditions and plotted to different y-axis due to the differences in the concentration ranges of the different sample subsets. Correlation of *MdTCP*, *SAP11CaPm* expression and phytoplasma levels (F) in graft-inoculated *Malus* × *domestica* leaf-samples from spring (grey, n = 3) and autumn (black, n = 4). Statistical analysis was performed with multiple t-test. Statistical differences were determined using the Holm-Sidak method, with alpha = 0.05 and linear regression analysis, using GraphPad Prism 7.05 (GraphPad Software Inc.). For pools comprising only one biological replicate, technical replicates accounted to the statistical analysis. Significant differences between groups are indicated with asterisks (* P≤0.05, ** P≤0.01, *** P≤0.001). Data points with very similar values might overlap in the graph and can thus misleadingly appear as a single data point. In only seven out of the ten samples from the greenhouse phytoplasma concentration could be determined (F). This was due to a lack of sample material for the DNA preparation which is necessary for phytoplasma detection.

**Table 1. Linear regression analysis of TCP expression depending on phytoplasma quantity or $SAP11_{CaPm}$ expression.** 95% confidence interval (CI), goodness of fit and significance level (alpha = 0.05) of linear regression, calculated with GraphPad Prism 7.05 (GraphPad Software Inc.), between normalized phytoplasma quantity and $SAP11_{CaPm}$, MdTCP16, MdTCP25 and MdTCP24 expression (left panel) and between normalized $SAP11_{CaPm}$ expression and MdTCPs expression in leaves from greenhouse (middle panel) and from foil-tunnel, respectively (right panel).

| | normalized phytoplasma quantity (greenhouse) | | | | normalized $SAP11_{CaPm}$ expression (greenhouse) | | | normalized $SAP11_{CaPm}$ expression (foil tunnel) | | |
|---|---|---|---|---|---|---|---|---|---|---|
| | $SAP11_{CaPm}$ | MdTCP16 | MdTCP25 | MdTCP24 | MdTCP16 | MdTCP25 | MdTCP24 | MdTCP16 | MdTCP25 | MdTCP24 |
| **95% CI** | | | | | | | | | | |
| Slope | -0.020 to 0.358 | 6.195 to 9.67 | -0.096 to -0.019 | -0.196 to 0.006 | 0.194 to 45.46 | -0.405 to 0.13 | -0.603 to 0.452 | -0.912 to 6.262 | -0.658 to -0.0958 | -0.320 to 0.562 |
| Y-intercept | 0.143 to 1.439 | -5.168 to 6.743 | 0.955 to 1.22 | 0.903 to 1.595 | -38.49 to 18.16 | 0.762 to 1.43 | 0.435 to 1.75 | -2.0 to 3.99 | 0.907 to 1.38 | 0.535 to 1.27 |
| X-intercept | -infinity to -0.498 | -0.995 to 0.585 | 12.01 to 54.6 | 7.031 to +infinity | -78.23 to 1.014 | 3.4 to +infinity | 2.708 to +infinity | -infinity to 0.456 | 1.923 to 10.28 | -infinity to -1.071 |
| | | | | | | | | | | |
| **Goodness of Fit** | | | | | | | | | | |
| $R^2$ | 0.513 | 0.965 | 0.743 | 0.540 | 0.403 | 0.150 | 0.0135 | 0.517 | 0.776 | 0.127 |
| | | | | | | | | | | |
| P value | 0.07 | <0.001 | 0.01 | 0.06 | 0.049 | 0.269 | 0.750 | 0.107 | 0.02 | 0.489 |
| Deviation from zero? | Not Significant | Significant | Significant | Not Significant | Significant | Not Significant | Not Significant | Not Significant | Significant | Not Significant |
| | | | | | | | | | | |
| **Equation** | Y = 0.169*X + 0.791 | Y = 7.932*X + 0.788 | Y = -0.057*X + 1.087 | Y = -0.095*X + 1.249 | Y = 22.83*X- 10.16 | Y = -0.1377*X + 1.097 | Y = -0.07555*X + 1.094 | Y = 2.675*X + 0.9928 | Y = -0.3771*X + 1.141 | Y = 0.1209*X + 0.9024 |
| | | | | | | | | | | |
| **Number of X values** | 7 | 7 | 7 | 7 | 10 | 10 | 10 | 16 | 16 | 16 |

95% confidence interval (CI), goodness of fit and significance level (alpha = 0.05) of linear regression, calculated with GraphPad Prism 7.05 (GraphPad Software Inc.), between normalized phytoplasma quantity and $SAP11_{CaPm}$, MdTCP16, MdTCP25 and MdTCP24 expression (left panel) and between normalized $SAP11_{CaPm}$ expression and MdTCPs expression in leaves from greenhouse (middle panel) and from foil-tunnel, respectively (right panel).

natural host plant and to understand the induced changes in TCP expression in the native pathosystem.

Our data show unequivocally that SAP11CaPm interacts with MdTCP16, both yeast and in planta, whereas another candidate (MdNYC1) does not interact (Figs 1 and 2). SAP11CaPm also binds to MdTCP25 and MdTCP24 from *Malus × domestica* and degrades different AtTCPs at the protein level, among them the orthologue of MdTCP25, i.e. AtTCP4 [16], but it is not known, whether binding of SAP11CaPm affects the stability of MdTCP16, since it was not able to destabilize the *Arabidopsis thaliana* orthologue AtTCP18 [11,31]. Interestingly, our study indicates that *MdTCP16* expression is induced during phytoplasma infection and might then be counter-regulated by SAP11CaPm at the protein level. A degradation of MdTCP16 induced by SAP11CaPm at the protein level could induce *MdTCP16* expression on the transcriptional level via a gene-product mediated feedback loop regulation [42]. The MdTCP16 protein might act like a repressor and negatively affects expression of the *MdTCP16* gene, meaning that skimming the proteinaceous gene product leads to an increase of its gene expression. That might explain why *MdTCP16* transcriptional expression levels are by trend increased in infected plants and that phytoplasma level in leaves is positively correlated with the expression of this transcription factor (Fig 3A–3E) which has been shown also for 'Ca. P. ziziphi' infected *Ziziphus jujube*, another phytoplasma pathosystem where the *ZjTCP7* that

encodes a MdTCP16 orthologue of *Z. jujube*, is upregulated in leaves [43]. It is worth noting that–in contrast to *MdTCP16* –*MdTCP25* expression is adversely i.e., negatively and *MdTCP24* expression not at all correlated with SAP11$_{CaPm}$ expression during infection. This implies that other regulatory mechanisms might be involved in the regulation of *MdTCP25* and *MdTCP24* expression.

Whatever pathway is affected during *MdTCP16* expression, it seems reasonable to speculate that it is important for the phytoplasma to attack MdTCP16 in its plant host.

The increased *MdTCP16* expression in autumn samples might be an indication that this TCP is involved in the seasonal control of branching. *MdTCP16* is mainly expressed in axillary and flower buds and only weakly in leaves, stems and shoot tips of apple trees [44]. Short photoperiods lead to the expression of *BRC1* –an orthologue of *MdTCP16* –controlling a complex network that regulates bud dormancy [45]. Interestingly, early bud break is a symptom of Phytoplasma-infected apple trees. It is tempting to speculate that 'Ca. P. mali' infection affects *MdTCP16* expression via SAP11$_{CaPm}$ in axillary buds and is thus involved in the induction of early bud break. Our study was, however, focused on gene expression analyses in leaves, thus based on the current data availability it would be very speculative to hypothesize about SAP11-$_{CaPm}$'s function in other plant tissues.

No SAP11$_{CaPm}$ expression was detected in leaf samples from naturally infected trees in spring, but the effector was detectable in leaves from greenhouse trees already in spring. This discrepancy might be explained by differences in environmental and physiological circumstances between the fully grown naturally infected trees and the small and young trees from the greenhouse. It is possible that the phytoplasma colonization of the canopy occurs earlier or that the bacterial colonization is more uniform in the foliage part of the smaller greenhouse plants.

'Ca. P. mali' infection leads to increased soluble sugar content in phloem sap [46] and in leaves [47]. Sugar promotes the bud outgrowth by acting as a repression-signal for *BRC1* expression [25,48]. This is neither in line with the increased *MdTCP16* expression that we find in phytoplasma infected apple leaves, nor with what has been described for *ZjTCP7* expression in phytoplasma infected Chinese jujube [43]. Thus, other factors might outcompete the repressing effect of increased sugar levels on *MdTCP16* expression, such as an increased auxin level in leaves of infected apple trees [49]. Auxin induces the *MdTCP16* expression but blocks the axillary bud outgrowth in non-infected plants [50,51].

Taken together, BRC1 and its homologues seem to be important molecular targets of SAP11-like effector proteins from different phytoplasma species. The results of this study prove the interaction of SAP11$_{CaPm}$ and the MdTCP16 transcription factor. SAP11-like proteins might be crucial for the successful phytoplasma colonization of the canopy in spring by downshifting BRC1. In the context of the current knowledge, it can be assumed that this downregulation is involved in the formation of lateral shoot outgrowth and early bud break, which are typical symptoms of 'Ca. P. mali' infection, and eponymous to the diseases name "apple proliferation" [31]. However, the factors involved in *BRC1*/*MdTCP16*/*AtTCP18* gene regulation are not easy to detangle, since they seem to be species- and tissue- dependent and regulated in a complex manner. By expressing SAP11-like proteins, phytoplasma target different members of the plant TCP family which serve as molecular hubs, to manipulate their plant hosts in a very sophisticated -but not yet fully understood- manner.

## Supporting information

**S1 Fig. Reference Sequence of XM_008376500.2.** The sequence includes the CDS for XP_008374722.1 (*MdTCP16*) with TCP domain and the identified part in the Y2H screen. (PDF)

**S2 Fig. Phytoplasma concentration of leaf and root samples.** The normalized phytoplasma concentration is given as the ratio of the 'Ca. P. mali' specific *16S* gene copies and the *Malus x domestica* single-copy gene *ACO*. Phytoplasma concentration was quantified in seven infected leaf samples from greenhouse plants (three from spring and four from autumn), in three naturally infected pooled leaf samples (one from spring, two from autumn) and in one naturally infected pooled root sample (autumn).
(PDF)

**S1 Table. Primers used in this study.** Lowercase letters indicate bases for Gateway-*attB* site overhangs or *Sfi*I restriction site overhangs.
(PDF)

# Acknowledgments

We would like to thank Christine Kerschbamer for lab assistance and Mirko Moser (Fondazione Edmund Mach, San Michele All'Adige, Italy) for discussing the data. Massimiliano Trenti, Erika Corretto from the Free University of Bolzano (Italy) and Pier Luigi Bianchedi from Fondazione Edmund Mach (San Michele all'Adige, Italy) for experimental assistance in grafting and sampling and Cameron Cullinan for English proofreading.

# Author Contributions

**Conceptualization:** Cecilia Mittelberger, Katrin Janik.

**Formal analysis:** Cecilia Mittelberger, Bettina Hause, Katrin Janik.

**Funding acquisition:** Katrin Janik.

**Investigation:** Cecilia Mittelberger, Katrin Janik.

**Methodology:** Cecilia Mittelberger, Bettina Hause, Katrin Janik.

**Supervision:** Bettina Hause, Katrin Janik.

**Writing – original draft:** Cecilia Mittelberger, Bettina Hause, Katrin Janik.

**Writing – review & editing:** Cecilia Mittelberger, Bettina Hause, Katrin Janik.

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
