## [Decision Letter · Decision Letter 0]

19 Sep 2022

PONE-D-22-20188The ‘Candidatus Phytoplasma mali’ effector protein SAP11CaPm interacts with MdTCP16, a class II CYC/TB1 transcription factor that is highly expressed during phytoplasma infection.PLOS ONE

Dear Dr. Janik,

Thank you for submitting your manuscript to PLOS ONE. After careful consideration, we feel that it has merit but does not fully meet PLOS ONE’s publication criteria as it currently stands. Therefore, we invite you to submit a revised version of the manuscript that addresses the points raised during the review process. The current submission does not meet PLOS ONE publication criteria primarily in areas of a sufficient description of the experimental materials and methods used, some issues with analyses of the data, and that significant conclusions are not adequately supported by the data provided. All of the reviewers' comments are relevant and should be addressed in a revised manuscript. It is particularly important that you address Reviewer 2's concerns about the interpretation of MdTCP16 expression. 

We look forward to receiving your revised manuscript.

Kind regards,

Keith R. Davis

Academic Editor

PLOS ONE

Journal Requirements:

Reviewers' comments:

Reviewer's Responses to Questions

**Comments to the Author**

1. Is the manuscript technically sound, and do the data support the conclusions?

Reviewer #1: Yes

Reviewer #2: Partly

2. Has the statistical analysis been performed appropriately and rigorously? 

Reviewer #1: Yes

Reviewer #2: I Don't Know

3. Have the authors made all data underlying the findings in their manuscript fully available?

Reviewer #1: Yes

Reviewer #2: No

4. Is the manuscript presented in an intelligible fashion and written in standard English?

Reviewer #1: Yes

Reviewer #2: Yes

5. Review Comments to the Author

Reviewer #1: The MS “The ‘Candidatus Phytoplasma mali’ effector protein SAP11CaPm interacts with MdTCP16, a class II CYC/TB1 transcription factor that is highly expressed during phytoplasma infection.” (PONE-D-22-20188) is a very nice little story which deals with an interesting mechanism of effector protein interacting with transcription factors.

The MS is written very well and is also well structured. Nevertheless, I have some critical points which should be considered for revision. The M+M part have to be improved as I could not found the protocol for protoplastation and also the growth conditions of the apple trees (field and greenhouse) are missing. Which apple cultivars are used and did the authors used the same cultivar for grafting (rootstock and scion)? Which Ca. P mali accession did the authors used in the grafted trees? (virulent or avirulent?) Can the interaction of SAP11 and MdTCP16 be different if the tree is infected with a virulent or avirulent accession? What about the phytoplasma titre of the trees, should be shown as suppl. table. Fig. 2: YFP signal seems also be in the membrane. That this is not the case should be checked by colocalization.

Reviewer #2: In this study, Mittelberger et al. investigate the interactions of the Candidatus phytoplasma mali effector protein SAP11 with two host proteins: the transcription factor MdTCP16 and NYC1, a chloroplast reductase. The paper also analyses the expression of three different Malus domestica transcription factors (MdTCP16, MdTCP24 and MdTCP25) during phytoplasma infection over the plant vegetative season. The main focus of the study is improving the knowledge about SAP11 role in Ca. P. mali infection identifying new interactors and describing the expression dynamic of SAP11 and its host targets between spring and autumn.

The manuscript is generally well-written and easy to follow. The two approaches adopted to study SAP11 targets congruently identify MdTCP16 as a new interactor of the effector protein. However, some issues with the methods and analysis of the gene expression studies need to be addressed.

Below are more specific comments by section:

Title, Abstract and Introduction:

- The title appropriately summarizes the content and results of the paper.

- The abstract is detailed, concise, and appropriately condenses the manuscript content.

- The introduction provides enough information to understand the background and the relevance of the study.

Methods:

- A description of the different plant growing conditions would help understand and interpret the results. Since BRC1 are influenced by photoperiod and temperature fluctuations, it is important to know if the plants in greenhouse were maintained in controlled conditions or not. I suggest introducing one or more sections to describe the plant materials and the plant growth conditions in the greenhouse (light, temperature). The sampling strategies, and the sampling size for naturally infected and greenhouse plants need also to be provided to better interpret the results and statistical analysis. See also comment to lines 221-224.

Results

- Line 193: NYC1 and TCP16 do not share common features, do not belong to the same protein family and are localized in different subcellular compartments. The co-expression of SAP11 and MdNYC1 further confirms the lack of interaction between these two proteins already suggested by the Y2H experiment. Therefore, I would not consider this result as the negative control but part of the experiment, since the BiFC experiment was conducted to validate the results obtained by the Y2H. Moreover,

- Line 221-224: the discrepancy mentioned in line 223 seems to refer to greenhouse vs naturally infected plants. Nevertheless, the authors suggested that the discrepancy “might be due to the season-dependent phytoplasma colonization”. This statement needs to be detailed and moved to the Discussion paragraph.

- Figure 3 Panel A and B. The plots show no difference in MdTCP16 expression between healthy and infected plants in spring and autumn. The fact that MdTCP16 expression increases during the season, such as the phytoplasma titer (panel D), is not enough to prove a direct correlation between the two phenomena. Moreover, the fact that MdTCP16 has the same trend of expression in healthy and infected plants shows that the increase of MdTCP16 transcription is correlated to environmental or plant physiological conditions rather than phytoplasma infection.

Panels C and D. It is unclear what samples are plotted here. The size of the data set differs between the two panels. It would be important to report and describe the data sets used to build the plots.

- Lines 251-252 See comment to Figure 3.

Discussion

- Line 286-290 The authors suggest that TCP16 might play a role in the early bud break during the spring. Nevertheless, in springtime TCP16 expression level is the lowest registered during the vegetative season. Moreover, in spring, the expression level of SAP11 was not detectable in the open field and was extremely low in the greenhouse.

A further concern is that the expression analysis of all the genes involved in the study was conducted on leaf samples, but the MdTCP16 is mainly expressed in buds, where it is supposed to play a crucial role. Similarly, the expression of SAP11 in bud cells was not investigated. Consequently, the hypothesis of the involvement of SAP11-TCP16 interaction in the induction of early bud break of phytoplasma infected trees is speculative and not supported by the data presented.

6. PLOS authors have the option to publish the peer review history of their article (what does this mean?). If published, this will include your full peer review and any attached files.

Reviewer #1: **Yes: **Alexandra C. U. Furch

Reviewer #2: No

---

## [Author Response · Author response to Decision Letter 0]

6 Oct 2022

Reviewer #1: 

The MS “The ‘Candidatus Phytoplasma mali’ effector protein SAP11CaPm interacts with MdTCP16, a class II CYC/TB1 transcription factor that is highly expressed during phytoplasma infection.” (PONE-D-22-20188) is a very nice little story which deals with an interesting mechanism of effector protein interacting with transcription factors. The MS is written very well and is also well structured. Nevertheless, I have some critical points which should be considered for revision. 

The M+M part have to be improved as I could not found the protocol for protoplastation and also the growth conditions of the apple trees (field and greenhouse) are missing. 

>>> Comment R1.1.: We introduced a Plant Material subheading in the M+M section, for describing the plant material. Additionally, the procedure for protoplast extraction and transformation is now described in detail.

Which apple cultivars are used and did the authors used the same cultivar for grafting (rootstock and scion)? 

>>> Comment R1.2.: The same cultivar was used. The details are now described in the newly added Plant Material subheading of the M+M section.

Which Ca. P mali accession did the authors used in the grafted trees? (virulent or avirulent?) 

>>> Comment R1.3.: The trees in the greenhouse have been grafted with the virulent strain PM6, an Italian AT2-subtype isolate derived from a naturally infected tree of cv. Golden Delicious characterized in Bisognin et al. 2007. This information has been added to the respective part in the M+M section. The other trees are infected with a local AT2 strain as determined by Sanger sequencing. Since these trees have not been infected with a pre-cultured pure phytoplasma strain it cannot be ruled out that a mixed infection with other strains is present in these trees. However, this situation resembles the actual one in in the field. 

Can the interaction of SAP11 and MdTCP16 be different if the tree is infected with a virulent or avirulent accession? 

>>> Comment R1.4.: This is an interesting question. In a previous analysis, variants of SAP11 from different AP strains in South Tyrol have been analyzed (Janik et al. 2017). No SAP11 variants on amino acid level could be identified in this study, leading to the conclusion that SAP11 is highly conserved in 'Ca. P. mali'. To our knowledge there is no report of 'Ca. P. mali' strains that have an aberrant or dysfunctional SAP11 protein or are completely lacking this protein. The role of SAP11 in avirulent and virulent 'Ca. P. mali' strains has not been analyzed, yet. Based on our data we cannot deduce a hypothesis whether SAP11 and MdTCP16 interaction is differing in virulent and avirulent 'Ca. P. mali' strains. It would be indeed interesting to further analyze the function of this effector in this direction in a future study.

What about the phytoplasma titre of the trees, should be shown as suppl. table. 

>>> Comment R1.5.: A supplementary table showing the phytoplasma quantification in the roots of the trees from the foil-tunnel used in this study has been added as Table S2. Unfortunately, a phytoplasma quantification in the roots of the small greenhouse trees for each time point has not been performed in parallel to the leaf analysis. However, data about the phytoplasma concentration in leaves is shown in Fig 3D.

Fig. 2: YFP signal seems also be in the membrane. That this is not the case should be checked by colocalization.

>>> Comment R1.6.: We apologize if the pictures that we provided are not fully unambiguous. A cytoplasm localization can sometimes appear as a localization close to the membrane, especially if it is very faint and the enlargement of cells is not high. Comparing the signals coming from YFP and RFP, it is visible that both perfectly co-localize. Free RFP of the transformation control is cytosolic and – due to its small size – also translocated to the nucleus. The MdTCP16-SAP11CaPm-complex (YFP) is also nuclear and cytoplasmic localized and we could not find any indication for a membrane localization of it. To avoid a potential misunderstanding for the readers of the article we increased the contrast in the image to improve the visual subcellular localization.

Reviewer #2: 

In this study, Mittelberger et al. investigate the interactions of the Candidatus phytoplasma mali effector protein SAP11 with two host proteins: the transcription factor MdTCP16 and NYC1, a chloroplast reductase. The paper also analyses the expression of three different Malus domestica transcription factors (MdTCP16, MdTCP24 and MdTCP25) during phytoplasma infection over the plant vegetative season. The main focus of the study is improving the knowledge about SAP11 role in Ca. P. mali infection identifying new interactors and describing the expression dynamic of SAP11 and its host targets between spring and autumn.

The manuscript is generally well-written and easy to follow. The two approaches adopted to study SAP11 targets congruently identify MdTCP16 as a new interactor of the effector protein. However, some issues with the methods and analysis of the gene expression studies 

need to be addressed.

Below are more specific comments by section:

Title, Abstract and Introduction:

-The title appropriately summarizes the content and results of the paper.

-The abstract is detailed, concise, and appropriately condenses the manuscript content.

-The introduction provides enough information to understand the background and the relevance of the study.

Methods:

- A description of the different plant growing conditions would help understand and interpret the results. Since BRC1 are influenced by photoperiod and temperature fluctuations, it is important to know if the plants in greenhouse were maintained in controlled conditions or not. I suggest introducing one or more sections to describe the plant materials and the plant growth conditions in the greenhouse (light, temperature). The sampling strategies, and the sampling size for naturally infected and greenhouse plants need also to be provided to better interpret the results and statistical analysis. See also comment to lines 221-224.

>>> Comment R2.1.: A new subheading was now introduced to the M+M section for description of plant growth conditions. Additionally, the sampling strategies and sampling size were completed with all information.

Results

- Line 193: NYC1 and TCP16 do not share common features, do not belong to the same protein family and are localized in different subcellular compartments. The co-expression of SAP11 and MdNYC1 further confirms the lack of interaction between these two proteins already suggested by the Y2H experiment. Therefore, I would not consider this result as the negative control but part of the experiment, since the BiFC experiment was conducted to validate the results obtained by the Y2H.

>>> Comment R2.2.: The wording was changed accordingly and the BiFC experiment considered as a confirmation of the Y2H results.

Moreover,

- Line 221-224: the discrepancy mentioned in line 223 seems to refer to greenhouse vs naturally infected plants. Nevertheless, the authors suggested that the discrepancy “might be due to the season-dependent phytoplasma colonization”. This statement needs to be detailed and moved to the Discussion paragraph.

>>> Comment R2.3.: The statement was moved to the discussion section and is now explained in more detail.

- Figure 3 Panel A and B. The plots show no difference in MdTCP16 expression between healthy and infected plants in spring and autumn. The fact that MdTCP16 expression increases during the season, such as the phytoplasma titer (panel D), is not enough to prove a direct correlation between the two phenomena. Moreover, the fact that MdTCP16 has the same trend of expression in healthy and infected plants shows that the increase of MdTCP16 transcription is correlated to environmental or plant physiological conditions rather than phytoplasma infection.

Panels C and D. It is unclear what samples are plotted here. The size of the data set differs between the two panels. It would be important to report and describe the data sets used to build the plots.

- Lines 251-252 See comment to Figure 3.

>>> Comment R2.4.: Figure 3 and its legend were changed according to the reviewer’s suggestion. The number of samples is now described in detail in the legend. You are right, that in Panel B there is no significant difference between MdTCP16 expression in non-infected and infected leaves. The rationale for the conclusion that MdTCP16 expression correlates with phytoplasma presence is based on the data depicted in panel D of this figure, in which a strong correlation between MdTCP16 and phytoplasma concentration can be observed. In panel A and B of this figure there is a trend that MdTCP16 is more expressed in infected leaves in autumn, especially in the (smaller) greenhouse trees. However, the difference is neither significant in the group of the greenhouse trees nor in the naturally infected trees. This initial observation (i.e. that there is a trend in autumn) and the fact that phytoplasma concentration in leaves is the highest in leaves in autumn led us to the question if there is a positive correlation between MdTCP16 expression and phytoplasma concentration in the leaves. The results of the correlation analysis are shown in panel D, where samples from greenhouse and naturally infected trees were analyzed together. In this evaluation a positive correlation between phytoplasma concentration and MdTCP16 expression could be observed. 

Discussion

- Line 286-290 The authors suggest that TCP16 might play a role in the early bud break during the spring. Nevertheless, in springtime TCP16 expression level is the lowest registered during the vegetative season. Moreover, in spring, the expression level of SAP11 was not detectable in the open field and was extremely low in the greenhouse.

A further concern is that the expression analysis of all the genes involved in the study was conducted on leaf samples, but the MdTCP16 is mainly expressed in buds, where it is supposed to play a crucial role. Similarly, the expression of SAP11 in bud cells was not investigated. Consequently, the hypothesis of the involvement of SAP11-TCP16 interaction in the induction of early bud break of phytoplasma infected trees is speculative and not supported by the data presented.

>>> Comment R2.5.: We agree and changed the discussion part accordingly and reduced the speculative part regarding an involvement of SAP11 in MdTCP16-mediated induction of early bud-break.

---

## [Decision Letter · Decision Letter 1]

25 Oct 2022

PONE-D-22-20188R1

The ‘Candidatus Phytoplasma mali’ effector protein SAP11CaPm interacts with MdTCP16, a class II CYC/TB1 transcription factor that is highly expressed during phytoplasma infection.

PLOS ONE

Dear Dr. Janik,

Thank you for submitting your manuscript to PLOS ONE. After careful consideration, we feel that it has merit but does not fully meet PLOS ONE’s publication criteria as it currently stands. Therefore, we invite you to submit a revised version of the manuscript that addresses the points raised during the review process.

We look forward to receiving your revised manuscript.

Kind regards,

Keith R. Davis

Academic Editor

PLOS ONE

Journal Requirements:

Additional Editor Comments:

Although you did a nice job of addressing many of the reviewers' concerns, Reviewer 2 still has some legitimate questions about the data analysis presented in Figure 3. I agree that combining data from treatments as currently presented is questionable and needs to be addressed. Once this is adequately done, the manuscript can be recommended for publication.

Reviewers' comments:

Reviewer's Responses to Questions

**Comments to the Author**

1. If the authors have adequately addressed your comments raised in a previous round of review and you feel that this manuscript is now acceptable for publication, you may indicate that here to bypass the “Comments to the Author” section, enter your conflict of interest statement in the “Confidential to Editor” section, and submit your "Accept" recommendation.

Reviewer #1: All comments have been addressed

Reviewer #2: (No Response)

2. Is the manuscript technically sound, and do the data support the conclusions?

Reviewer #1: Yes

Reviewer #2: Partly

3. Has the statistical analysis been performed appropriately and rigorously? 

Reviewer #1: Yes

Reviewer #2: Yes

4. Have the authors made all data underlying the findings in their manuscript fully available?

Reviewer #1: Yes

Reviewer #2: Yes

5. Is the manuscript presented in an intelligible fashion and written in standard English?

Reviewer #1: Yes

Reviewer #2: Yes

6. Review Comments to the Author

Reviewer #1: Dear authors,

Thank you very much for the revised version. All my raised comments were adequately addressed.

Good luck for your future work.

Reviewer #2: The authors have clarified most of the issues I raised during my last review. I still have a few points about Figure 3 that need to be addressed:

Figure 3C: Since the samples plotted here come from different growing conditions (naturally vs graft infected and greenhouse vs foil tunnel) and, as shown in panels A and B, have different expression profiles, they shouldn't be analyzed as a single set of samples. The expression level of MdTCP16 in grafted plants is more than ten times lower than the transcription level in naturally infected trees. Similarly, the correlation analysis should be performed separately.

Figure 3D: Five grafted inoculated samples were plotted in figure 3B, but in figure 3D only three samples for spring and four for autumn were plotted. What is the rationale behind the exclusion of some of the samples?

7. PLOS authors have the option to publish the peer review history of their article (what does this mean?). If published, this will include your full peer review and any attached files.

Reviewer #1: No

Reviewer #2: No

---

## [Author Response · Author response to Decision Letter 1]

2 Nov 2022

Subject: Resubmission of PONE-D-22-20188R1

thank you very much for the valuable input and the comments regarding the previous version of our manuscript with the title “The 'Candidatus Phytoplasma mali' effector protein SAP11CaPm interacts with MdTCP16, a class II CYC/TB1 transcription factor that is highly expressed during phytoplasma infection”. Along the revised manuscript we provide point-by-point responses to every reviewer’s concern. Every author’s response to a reviewer’s comment is indicated with arrows (>>>). All changes that have been performed to the previous version of the manuscript are indicated by tracked changes. The manuscript text, figures, legends, or any other part of the previously submitted documents have not undergone any substantial changes if not requested by the reviewers. 

We hope that our manuscript now fulfills all criteria for publication in PLOS ONE. 

Reviewer #1: Dear authors,

Thank you very much for the revised version. All my raised comments were adequately addressed.

Good luck for your future work.

Reviewer #2: The authors have clarified most of the issues I raised during my last review. I still have a few points about Figure 3 that need to be addressed:

Figure 3C: Since the samples plotted here come from different growing conditions (naturally vs graft infected and greenhouse vs foil tunnel) and, as shown in panels A and B, have different expression profiles, they shouldn't be analyzed as a single set of samples. The expression level of MdTCP16 in grafted plants is more than ten times lower than the transcription level in naturally infected trees. Similarly, the correlation analysis should be performed separately.

>>> The correlation analyses shown in the former Fig 3C and Table 1 were changed according to the reviewer’s suggestions. This improved the comprehensibility of the whole graph and thus the interpretation of the results. Expression levels were compared separately and regarding the growing conditions. The statistical correlation analysis was also performed on the separated sample sets as suggested. The adjusted analytical approach did not affect the overall results, i.e. the same results or trends were found in the Figures 3 C-D as in the former Fig 3C. The text in the manuscript was changed according to the changes in the analyses and the graph.

Figure 3D: Five grafted inoculated samples were plotted in figure 3B, but in figure 3D only three samples for spring and four for autumn were plotted. What is the rationale behind the exclusion of some of the samples?

>>> For the phytoplasma quantification in Fig 3D DNA was required whereas for the expression analyses RNA/cDNA was used. Since the phytoplasma concentration was determined in a second moment there was not enough material for a DNA extraction for every sample left. We tried a phytoplasma quantification based on cDNA but did not succeed. The comparison was performed as follows: In a pilot-study with a subset of samples we compared the phytoplasma quantification results using cDNA with those using genomic DNA from the same sample material. Summarizing the results that we got, we found that DNA-based quantification results did not correspond to those based on cDNA performed in parallel. We must admit that we did not further invest time in finding an appropriate phytoplasmal housekeeping gene that could have been used for accurate cDNA-based phytoplasma quantification. That’s why we -unfortunately- cannot show results all ten samples in Fig 3D. To avoid appearing untransparent in this regard, we now mention the reason for the discrepancy in greenhouse sample numbers between 3B and 3F (before 3D) in the figure legend.

---

## [Decision Letter · Decision Letter 2]

14 Nov 2022

The ‘Candidatus Phytoplasma mali’ effector protein SAP11CaPm interacts with MdTCP16, a class II CYC/TB1 transcription factor that is highly expressed during phytoplasma infection.

PONE-D-22-20188R2

Dear Dr. Janik,

We’re pleased to inform you that your manuscript has been judged scientifically suitable for publication and will be formally accepted for publication once it meets all outstanding technical requirements.

Kind regards,

Keith R. Davis

Academic Editor

PLOS ONE

Additional Editor Comments (optional):

Thank you for your patience as we worked our way through the reviews. I hope you agree that the manuscript is improved and more useful for the research community. Best of luck in your future studies.

Reviewers' comments:

Reviewer's Responses to Questions

**Comments to the Author**

1. If the authors have adequately addressed your comments raised in a previous round of review and you feel that this manuscript is now acceptable for publication, you may indicate that here to bypass the “Comments to the Author” section, enter your conflict of interest statement in the “Confidential to Editor” section, and submit your "Accept" recommendation.

Reviewer #2: All comments have been addressed

2. Is the manuscript technically sound, and do the data support the conclusions?

Reviewer #2: Yes

3. Has the statistical analysis been performed appropriately and rigorously? 

Reviewer #2: Yes

4. Have the authors made all data underlying the findings in their manuscript fully available?

Reviewer #2: Yes

5. Is the manuscript presented in an intelligible fashion and written in standard English?

Reviewer #2: Yes

6. Review Comments to the Author

Reviewer #2: The authors did a good job and addressed all my concerns.

Only one little comment below:

Lines 272-274: "Lines in graphs C, D, E and F show linear regression of the respective samples either from the greenhouse (dashed line) or from the foil tunnel (solid line)": In Figures 3C-E, the regression lines are all solid and dashed lines are missing

7. PLOS authors have the option to publish the peer review history of their article (what does this mean?). If published, this will include your full peer review and any attached files.

Reviewer #2: No

---

## [Editor Report · Acceptance letter]

7 Dec 2022

PONE-D-22-20188R2 

The ‘*Candidatus* Phytoplasma mali’ effector protein SAP11_CaPm_ interacts with MdTCP16, a class II CYC/TB1 transcription factor that is highly expressed during phytoplasma infection. 

Dear Dr. Janik:

I'm pleased to inform you that your manuscript has been deemed suitable for publication in PLOS ONE. Congratulations! Your manuscript is now with our production department. 

Kind regards, 

on behalf of

Dr. Keith R. Davis 

Academic Editor

PLOS ONE